

# Kicking it off(-shell) with direct diffusion

Anja Butter[1,2], Tomáš Ježo[3], Michael Klasen[3], Mathias Kuschick[3],
Sofia Palacios Schweitzer[1] and Tilman Plehn[1]

**1** Institut für Theoretische Physik, Universität Heidelberg, Germany
**2** LPNHE, Sorbonne Université, Université Paris Cité, CNRS/IN2P3, Paris, France
**3** Institut für Theoretische Physik, Universität Münster, Germany

## Abstract

Off-shell effects in large LHC backgrounds are crucial for precision predictions and, at the same time, challenging to simulate. We present a novel method to transform high-dimensional distributions based on a diffusion neural network and use it to generate a process with off-shell kinematics from the much simpler on-shell one. Applied to a toy example of top pair production at LO we show how our method generates off-shell configurations fast and precisely, while reproducing even challenging on-shell features.

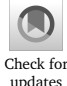

## 1 Introduction

Fast and precise theoretical predictions from first principles are required for essentially every experimental LHC analysis. This is especially true for modern inference methods, where the complete phase space coverage ensures an optimal measurement [1,2]. Based on perturbative quantum field theory, precision simulations of the hard scattering process face two challenges, the loop order and the multiplicity of the partonic final state [3]. The latter increases rapidly for example, when we describe the production of decaying heavy particles. Naively, one could expect that describing the decay kinematics close to the mass shell of the heavy particles, for

example using a Breit-Wigner propagator, is sufficient. However, given the precision targets of the upcoming LHC runs, on-shell approximations are no longer justified [4, 5]. In view of the HL-LHC, simulations need to describe heavy particle production and decay including, both, quantum corrections and off-shell kinematics.

In addition to the theoretical and computational effort behind precise calculations, the question is how they can be made available through multi-purpose event generators. Here we can resort to modern machine learning (ML) [1, 6]. Following the modular structure of event generators, the most obvious ML-applications are neural network surrogates for expensive scattering amplitudes [7–13]. Aside from the speed, these surrogates have the advantage that they can always be evaluated in parallel on GPUs [14–17]. Given these amplitudes, we can then improve the phase-space integration and sampling [18–26]. One advantage is that simulations are defined over interpretable physics phase spaces, for example scattering events [27–33], parton showers [34–41], and detector simulations [42–65]. All of these speed improvements maintain the first-principles nature of LHC event generators, which means they take theory predictions and evaluate them using faster ML-methods.

The goal of this study is to develop a precise and fast generative network that maps two LHC phase spaces onto each other [65–67]. If one phase space is a narrow sub-manifold of the other one, like in the case of on-shell and off-shell phase spaces of production of heavy unstable particles, one cannot use regression or classifier reweighting but must resort to generative networks. For LHC applications, we already know that precision-generative networks [32, 68–71] are easy to ship and powerful in amplifying their training data [72, 73]. Tasks we can solve with their help include event subtraction [74], event unweighting [75, 76], or super-resolution enhancement [77, 78]. Their conditional versions enable new analysis methods, like probabilistic unfolding [67, 79–87], inference [88–90], or anomaly detection [91–96]. Currently, the best-performing network architectures are normalizing flows for invertible tasks and diffusion networks for sampling problems, with help from transformers for combinatorics.

In this paper we present such a mapping and use it to efficiently generate events over an off-shell phase space from given on-shell events. The idea behind this sampling from on-shell events is that the generative network does not have to reproduce the on-shell features and can focus on the additional and relatively smooth off-shell extension. We start by describing our training dataset and the problem of off-shell event generation in Sec. 2. Our generative network setup, based on a Bayesian- Conditional Flow Matching (CFM) architecture [33], is presented in Sec. 3. To control the generative network performance [71] and to improve the precision of the kinematic distributions [32], we apply a classifier reweighting in Sec. 4. We present an Outlook in Sec. 5.

## 2 Off-shell vs. on-shell events

Our benchmark process is the complete off-shell top pair production followed by leptonic decays,

$$pp \to b e^+ \nu_e \, \bar{b} \mu^- \nu_\mu \,. \tag{1}$$

Top pair production with its rich resonance structure is known to challenge phase space sampling, just as generative networks for transition amplitudes [30].

In the factorized approach, in which the production and decay processes decouple, this process is known all the way up to NNLO-QCD [97–100], NLO-EW [101–105] and NNLO-QCD combined with NLO-EW [106] in the production process; NNLO-QCD [107, 108] in the decay; and NNLO-QCD in both production and decay [109–111]. Full off-shell calculations in the dilepton channel are so far only available at NLO-QCD [112–117], but a calculation of full off-shell top pair production in association with an extra jet at NLO is also available in

Ref. [118]. In this pilot study we restrict ourselves to leading order in QCD, $\mathcal{O}(\alpha_S^2 \alpha^4)$, and reserve the application of our method to higher-order predictions to a future publication.

We generate event samples for 13 TeV proton-proton collisions with NNPDF31_nlo_as_0118 parton distributions [119]. The neutrinos, charged leptons and quarks of the first two generations are treated as massless, and the CKM matrix is assumed to be trivial. All input parameters are given in Tab. 1, and the electromagnetic coupling $\alpha$ and the weak mixing angle are derived from the weak gauge-boson masses and the Fermi constant.

Our study and our results can be extended to higher-order predictions or other processes by combining different jet multiplicities in the final state [90]. Our two benchmark datasets are generated with HVQ [120] and BB4L [121,122], respectively. The data generated with HVQ only includes approximate off-shell effects using a finite top width and including spin correlations [123] and is referred to as on-shell data. The generator is based on the POWHEG method [124,125] and is part of the POWHEG BOX V2 [126] package.

The data generated with BB4L takes into account full off-shell effects also including singly-resonant and non-resonant contributions and the corresponding interferences. The generator employs the POWHEGRES method [127], tailored for simulations with unstable particles. In this case, $W$ bosons and $b$ quarks do not always stem from a top decay.

In Fig. 1 we illustrate the size of the off-shell effects for a selection of kinematic distributions of final-state leptons and $b$-quarks. These distributions are not meant to compare realistic predictions with different treatment of off-shell effects, but rather our two datasets, so no event selection criteria are applied. For the invariant mass of the lepton-$b$ system we ensure, through charge identification, that the two particles come from the same (anti)top decay. Correspondingly, their invariant mass has an upper edge that does not exist for off-shell events. For the reconstructed top mass, or the invariant mass of the three decay products, we clearly see the Breit-Wigner propagator form, with an explicit cutoff. Far below the actual top mass, the off-shell prediction develops a shoulder at the $W$-mass. The secondary panels show the ratios of the integrated one-dimensional phase space densities, illustrating that a reweighting strategy between the two samples is unlikely to work.

The strategy behind our surrogate network is to generate an off-shell event dataset and learn its structures relative to a corresponding on-shell event dataset. This strategy can be applied to any process and at any order in perturbation theory. Our two datasets consist of 5M unit-weight events each. The six particles in the final state of Eq.(1) are represented by $\{p_T, \eta, \phi\}$, with fixed external particle masses. We remove three degrees of freedom through a global azimuthal rotation and by enforcing two-dimensional transverse momentum conservation, leaving us with a 15-dimensional phase space. Each on-shell requirement replaces a full phase space dimension by a fixed range, given by the Breit-Wigner shape with a hard-coded cutoff in the reconstructed invariant mass. Moreover, we conceal information relevant for eventual parton showering like the colourflow configuration or resonance history assignment. This is appropriate because the BB4L generator in its default setup does not distinguish between the $t\bar{t}$ and single-top resonance histories and assigns colourflow correspondingly.

Table 1: Parameters used for the generation of the training datasets.

| $m_t$ | 172.5 GeV | $\Gamma_t$ | 1.453 GeV |
|---|---|---|---|
| $m_b$ | 4.75 GeV | | |
| $m_Z$ | 91.188 GeV | $\Gamma_Z$ | 2.441 GeV |
| $m_W$ | 80.419 GeV | $\Gamma_W$ | 2.048 GeV |
| $m_H$ | 125.0 GeV | $\Gamma_H$ | 0.0403 GeV |
| $\mathcal{B}(W \to e\nu/\mu\nu)$ | | 1/9 | |
| $G_F$ | | $1.16585 \times 10^{-5} \text{GeV}^{-2}$ | |

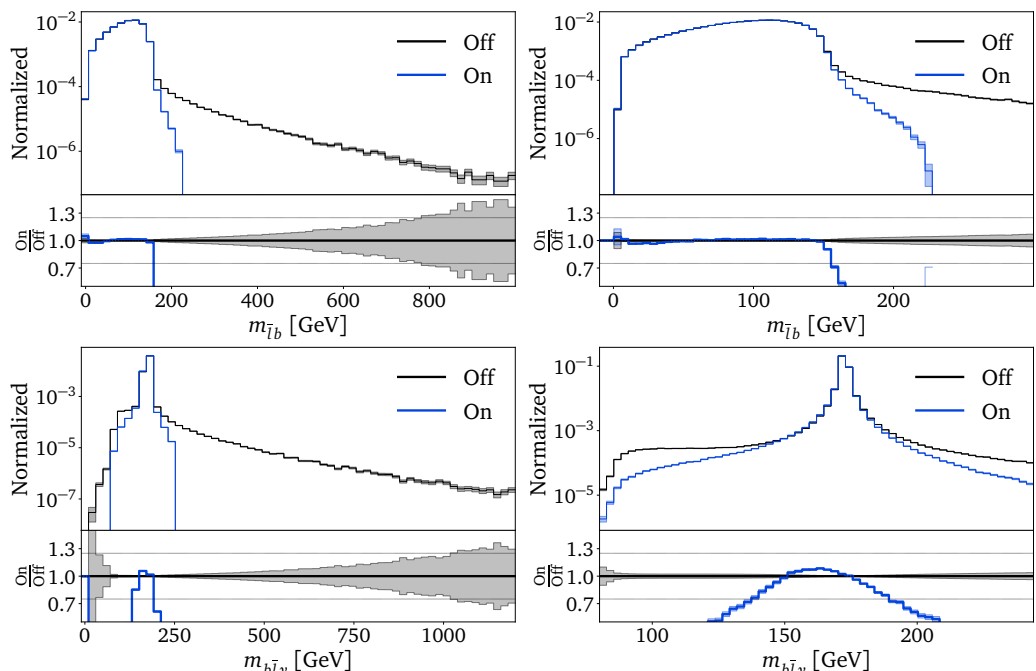

Figure 1: Example distributions for on-shell and off-shell $t\bar{t}$ event samples, illustrating the different phase space coverage. The right panels show the same distribution as the left panels, but zoomed into the respective bulk region.

For the network input, the kinematic variables are preprocessed: we scale the transverse momenta to $p_T^{1/3}$ and express the azimuthal angles as $\text{arctanh}(\phi/\pi)$. As subsequent classifier input, it turns out that $p_T^{-1/3}$ leads to the best results. Finally, all input dimensions are standardized to zero mean and unit variance.

## 3 Direct diffusion

For the network training, we start with the condition that we do not want to train on paired on-shell and off-shell events, because such a pairing does not follow a well-defined algorithm. This does not mean it is impossible to construct such a mapping, but we expect it to lead to artifacts. Instead, we will train a generative Direct Diffusion (DiDi) network on two phase space densities, one from on-shell and one from off-shell events.

Usually, we employ generative networks to learn a mapping between a simple latent space and some kind of phase space. The distribution in the latent space is sampled from, so we typically use a uniform or a Gaussian distribution. In this application, the latent space corresponds to the on-shell phase space, and the mapping is trained to generate off-shell events,

$$x \sim p_{\text{on}}(x) \quad \longleftrightarrow \quad x \sim p_{\text{model}}(x|\theta) \approx p_{\text{off}}(x). \tag{2}$$

Using the conditional flow matching (CFM) setup for LHC events [33, 128], we encode the transformation from on- to off-shell events as a continuous time evolution, which follows an ordinary differential equation (ODE)

$$\frac{dx(t)}{dt} \equiv v(x(t), t). \tag{3}$$

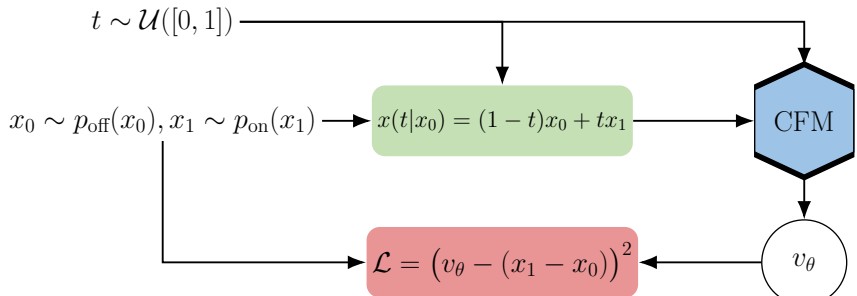

Figure 2: Training procedure for a CFM network mapping between on-shell and off-shell phase space distributions. Diagram adapted from Ref [33].

In terms of the related probability density, we can formulate the same task as

$$\frac{\partial p(x,t)}{\partial t} + \nabla_x \left[ p(x,t)v(x,t) \right] = 0.$$
(4)

The ODE and the continuity equation are equivalent, so we can train a network to represent the velocity field and use this velocity field to generate samples using a fast ODE solver for Eq.(3). Defining our time dependent probability density as

$$p(x,t) \to \begin{cases} p_{\text{off}}(x), & t \to 0, \\ p_{\text{on}}(x), & t \to 1, \end{cases}$$
(5)

we now need to construct the associated velocity field.

Unlike for the usual generative setup, we define $x(1) = x_1$ as a sample from the on-shell phase space, whereas $x(0) = x_0$ corresponds to the off-shell phase space. Therefore, we adapt the linear trajectory between on-shell and off-shell events to

$$x(t|x_0) = (1-t)x_0 + tx_1 \to \begin{cases} x_0, & t \to 0, \\ x_1 \sim p_{\text{on}}, & t \to 1. \end{cases}$$
(6)

The true conditional velocity field of our linear trajectory linked to our probability density in Eq. (5) is hence given by

$$v(x(t|x_0), t|x_0) = \frac{d}{dt} \left[ (1-t)x_0 + tx_1 \right] = -x_0 + x_1.$$
(7)

The remaining derivation is in complete analogy to Ref. [33], leading to the simple MSE loss

$$\mathcal{L}_{\text{CFM}} = \left\langle \left[ v_\theta((1-t)x_0 + tx_1, t) - (x_1 - x_0) \right]^2 \right\rangle_{t \sim \mathcal{U}([0,1]), x_0 \sim p_{\text{off}}, x_1 \sim p_{\text{on}}}.$$
(8)

Because the CFM setup does not include a likelihood, we can go directly from $p_{\text{on}}$ to $p_{\text{off}}$, without extra effort due to detours like in Flows4Flows [66], and without any pairing between $x_0 \sim p_{\text{off}}$ and $x_1 \sim p_{\text{on}}$.

As usual, we use a Bayesian version of the generative network [32, 129], to extract uncertainties on the learned phase space density. The Bayesian CFM loss [33]

$$\mathcal{L}_{\text{B-CFM}} = \left\langle \mathcal{L}_{\text{CFM}} \right\rangle_{\theta \sim q(\theta)} + c \, \text{KL}[q(\theta), p(\theta)],$$
(9)

includes a hyperparameter $c$ to balance the regular CFM loss with the Bayesian regularization term. If the first loss term was a likelihood loss, this factor would be fixed by Bayes' theorem. We have checked that the network performance is stable over many orders of magnitudes

for $c = 10^{-10} \dots 10^{-2}$. For larger values we observe that the training becomes unstable, as expected, while for very small values the uncertainty can no longer be captured. In addition, the prior weight distribution $q(\theta)$ is given by a unit Gaussian, where the choice of width hardly affects the network performance. Compared to the deterministic counterpart, this Bayesian network is equally precise.

The network and training setup is visualized in Fig. 2. We use a simple dense network with SiLU activation, where the last layer is initialized at zero. This sets the initial velocity field to zero and induces an identity mapping at the starting point of the training. We do not enforce a prescription for turning a given on-shell event into an off-shell event through the training data or the training procedure. In fact, for each epoch different phase space points will be connected via a linear trajectory. Instead, the network training constructs its own mapping to populate the off-shell phase space. This happens as part of the loss minimization, which means the transport map follows from an implicit measure encoded in the network architecture and loss.

As the CFM training objective is a simple regression, we can train on 17 dimensions, including two redundant degrees of freedom, as this happens to increase the precision. The two additional observables, the transverse momentum and the polar angle of the neutrino, are determined by transverse momentum conservation and are hence multidimensional correlations. Empirically we found that it is easier for the model to learn the behavior of those observables when handed directly. Especially more complicated correlations such as the invariant mass of the reconstructed top benefit greatly from this additional information. While improving the efficiency of the training these dimensions will be ignored for the actual event generation.

The network hyperparameters and the training parameters are given in Tab. 2. We encode $t$ in a higher embedding dimension and following [128] we add batch-wise random noise of scale $10^{-4}$ to $x_0$ and $x_1$ during training. We use the standard *Dopri5* ODE solver to sample from our network. In the interest of precision we use a large batch size. This is a problem for generic optimal-transport networks [130], but can be easily implemented in our architecture. We tested the OT-CFM [130] and found that in our setup the required small batch size indeed led to worse performance.

In Fig. 3 we show a set of one-dimensional kinematic distributions, for the on-shell data we start from, the off-shell training data, and the generated off-shell data. In the first panel we see how the network learns subtle differences almost perfectly well. The typical precision is around the per-cent level. For complex and sensitive correlations, like the lepton-$b$ invariant mass and the reconstructed top mass we start with a huge deviation between the on-shell distribution and the off-shell target. To generate these distributions correctly, the generative network has to learn correlations in the corresponding 9-dimensional sub phase space.

Table 2: Generative network setup (DiDi) and hyperparameters.

| Hyperparameter | |
| --- | --- |
| Embedding dimension | 64 |
| Layers | 8 |
| Intermediate dimensions | 768 |
| LR scheduling | OneCycle |
| Starter LR | $10^{-4}$ |
| Max LR | $10^{-3}$ |
| Epochs | 1000 |
| Batch size | 16384 |
| $c$ | $10^{-3}$ |
| # Training events | 3 M |

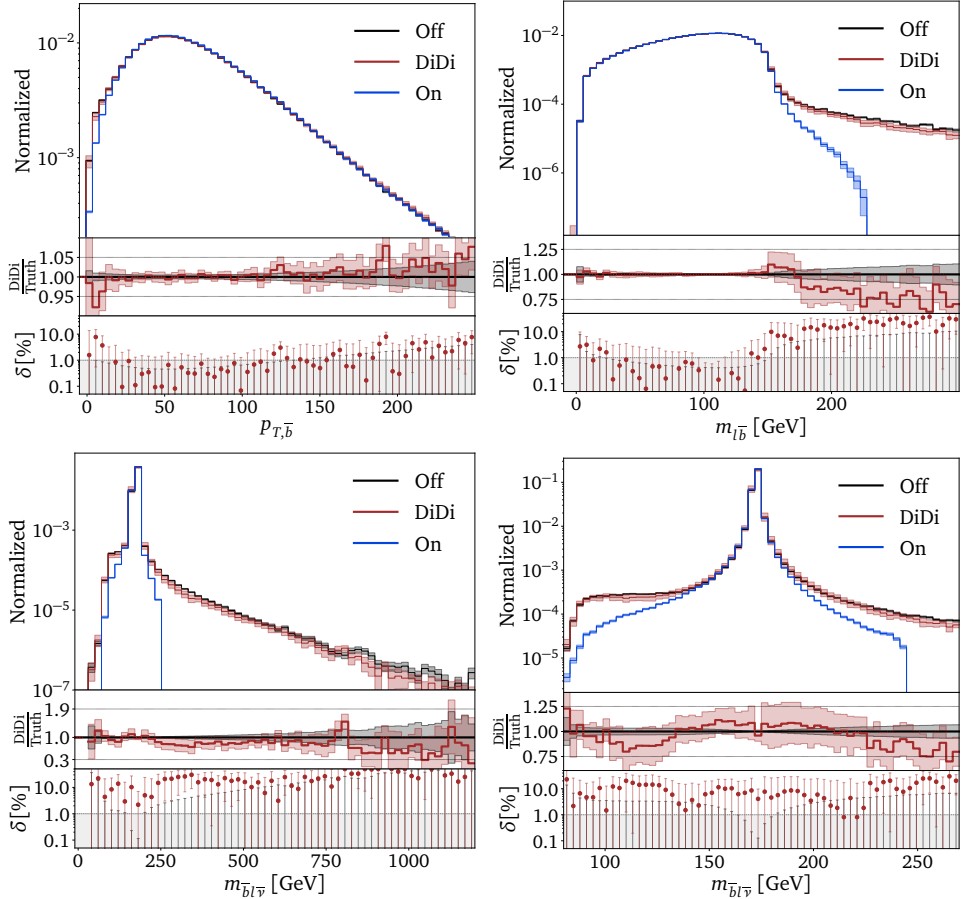

Figure 3: Results from our Direct Diffusion off-shell generator, compared to the on-shell starting point and the off-shell training distributions.

Our results confirm that in phase space regions covered well by both datasets, the generative network reproduces the target distribution precisely, as one could expect. However, we also see that even in phase space regions not populated by on-shell events the target distribution is reproduced relatively precisely. The typical agreement between the generated and target densities is around 10% in phase space regions with relatively little training data. While this deviation is covered by the uncertainties of the Bayesian network, we propose possible improvements in the next section.

Finally, we can ask how our generative network fills the off-shell phase space from the on-shell events. In Fig. 4 we show the correlations between the generated off-shell distribution and the on-shell starting distributions, i.e. the migration of paired latent and target phase space events from the forward simulation, where we emphasize that the pairing is only defined by the network evaluation, not by the training. In general, the correlation between the kinematic observables should be close to the identity, as confirmed by the $p_{T,\bar{b}}$ distribution in the left panel. This means that the shift from on-shell to off-shell phase space is relatively small and uncorrelated. However, for the reconstructed top mass, some of the events have to be shifted by a larger value, as illustrated in the right panel. While the width of the linear correlation becomes very small around the top mass peak, it rapidly increases away from the peak, demonstrating the large shift required to populate the off-shell phase space. Moreover, our network maps events from each side of the Breit-Wigner peak to the same side of the off-shell distribution.

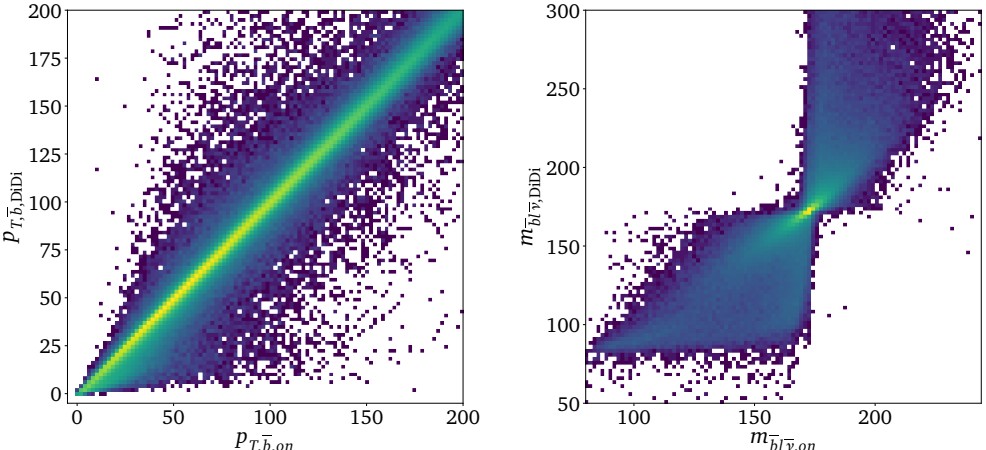

Figure 4: Migration plot — correlation between generated (off-shell) and starting (on-shell) distributions for two kinematic correlations. It illustrates the mapping found by the network during training.

## 4 Classifier control and reweighting

Because the unsupervised density estimation underlying our generative network is more challenging and less precise than a supervised classifier training, we can use a trained classifier as a function of phase space to systematically check and improve the generative network. A perfectly trained statistical classifier converges to the ratio of likelihoods,

$$C(x) = \frac{p_{\text{data}}(x)}{p_{\text{data}}(x) + p_{\text{model}}(x)} . \tag{10}$$

As a function of phase space we can use this classifier to construct an event reweighting, which improves the precision of the generative networks [32],

$$w(x) = \frac{p_{\text{data}}(x)}{p_{\text{model}}(x)} = \frac{C(x)}{1 - C(x)} . \tag{11}$$

In addition, we can use the same learned event weights to determine the precision of the generative network and systematically search for failure modes by searching for clusters of very small or very large weights in phase space [71].

We train the classifier on 27 observables, our 15 physical dimensions, complemented by the reconstructed top and anti-top masses, the reconstructed $W^+$ and $W^-$ masses, the reconstructed masses of the $\bar{b}l$ and $b\bar{l}$ systems, and the six corresponding transverse momenta. For the input of our classifier we sample events from our Bayesian generator setting each network weight to its mean value. The setup is given in Tab. 3.

We show histograms of learned phase space weights $w(x_i)$ on a linear and a logarithmic axis in Fig. 5. As expected, the distributions peak at unit weights, with a width around 0.3. Following Eq.(11), large weights correspond to phase space regions where the generative network produces a too small density of off-shell points; small weights mark phase space regions where the generative network produces too many off-shell events, compared to the training data. To study both tails of the weight distribution, we evaluate the weights over the combination of 2M training and 2M generated events and confirm that both tails decrease rapidly. We eventually clip the event weights to $w < 15$ to improve the numerical behavior of our generation and avoid sparks in regions of low statistics.

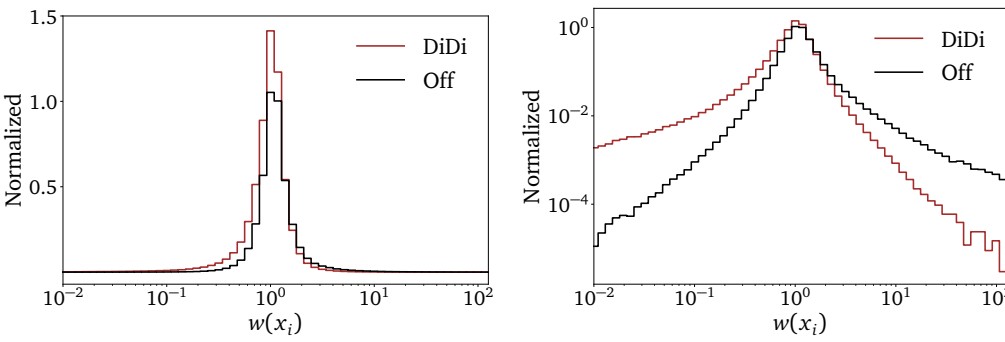

Figure 5: Histogram of the learned event weights, evaluated on the off-shell training data and the DiDi-generated off-shell events. The two panels show the same weights, on linear and logarithmic scales.

In Fig. 6 we track phase space regions with very small and very large weights. We compare all events of our test sample to the subsets with $w(x) < 0.6$, corresponding to 12.3% of the sample, and $w(x) > 1.6$, corresponding to 6.2% of the generated sample. The two shown distributions illustrate the general feature that the kinematic distributions for both tails are similar. In the $p_{T,\bar{b}}$-distribution we can identify two limitations of the network training: for small transverse momenta a slight shift of the cliff will lead to large relative weight corrections, while for large transverse momenta the decreasing density of training events will increase the relative size of the noise. For the reconstructed anti-top mass, one of the critical distributions for our generative task, the low-weight and high-weight tails are also comparable with each other and also comparable on both sides of the mass peak. The only distinct feature appears around the peak where events with small weight dominate the slightly over populated sides of the peak. This indicates that all weight tails are generated by noise, there are no missing localized features, and the limiting factor is the statistics of the training data.

Even though the shortcomings of our generative networks, visible in Fig. 3, arise from noisy network training and do not reflect systematic shortcomings, they affect the trained generative network in a systematic, localized manner. As a function of phase space, we can correct them using the event weights from Eq.(11), because the classifier network is more sensitive and more precise than the generator network [71]. In Fig. 7 we show a set of kinematic distributions for reweighted events, where the uncertainty is given by the Bayesian generator. Comparing the agreement between the reweighted and the target distribution with the unweighted performance from Fig. 3 we see significant improvements. In the secondary panels we show the

Table 3: Classifier network setup and hyperparameters.

| Hyperparameter | |
|---|---|
| Layers | 5 |
| Intermediate dimensions | 512 |
| Dropout | 0.1 |
| Normalization | BatchNorm1d |
| LR scheduling | ReduceOnPlateau |
| Starter LR | $1^{-3}$ |
| Patience | 10 |
| Epochs | 100 |
| Batch size | 1024 |
| # Training events | 2.5 M |

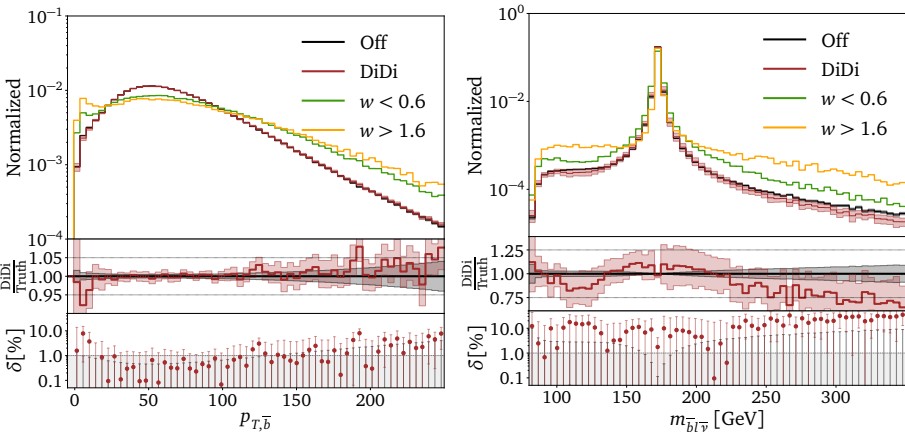

Figure 6: Clustering of the classifier trained to distinguish the off-shell training data from DiDi-generated events, for two example distributions.

reweighted predictions from the Bayesian generator, allowing us to compare the statistical uncertainty from the training data with the predictive uncertainty from the generative network. For the entire range of $p_{T,\bar{b}}$ the network agrees with the true distribution within its predictive uncertainties, and almost within the uncertainties of the training data. For $m_{\bar{l}b}$, DiDi has to cover phase space far beyond the on-shell structures, and, again it hardly exceeds the statistical uncertainties of the training data and provides a conservative uncertainty estimate from the Bayesian setup.

The momentum and the mass of the reconstructed $W^-$ match the truth perfectly in the bulk, and roughly within the statistics of the training data in the tails. The former is important, because it confirms the original motivation that the network reproduces the on-shell features with extremely high precision. The same can be seen for the reconstructed anti-top mass, where most of the phase space is filled by extrapolating from the on-shell distribution and the network learns these new phase space regions without degrading the precision in the bulk at all.

## 5 Outlook

Sufficiently fast and precise event generation for the HL-LHC is a major challenge in the coming decade. A perfect example is the generation of off-shell kinematic configurations for large backgrounds. Even when they can be computed from perturbative QFT, with significant investment in CPU time, they need to be implemented in standard event generators. A promising strategy to provide them in an amortized manner is generative surrogate networks. They are easy to include and ship with event generators and enhance the precision of a limited number of events, just like a fitted function to a statistically limited dataset.

Generating off-shell events is also an exciting problem for modern machine learning, because it cannot be solved by regressing an amplitude at a give phase space point. Instead, it requires a generative surrogate to cover the off-shell phase space. We proposed a new method, namely to generate off-shell configurations relative to given on-shell configurations. Its advantage is that the generative network only learns a controlled deviation from a simple unit transformation. This simplified task allows us to generate a rich resonance structure without the usual challenges in network size and precision training. The approach is enabled by Direct Diffusion (DiDi) networks, which sample from any given distribution to produce another distribution, in our case transforming on-shell events into off-shell events.

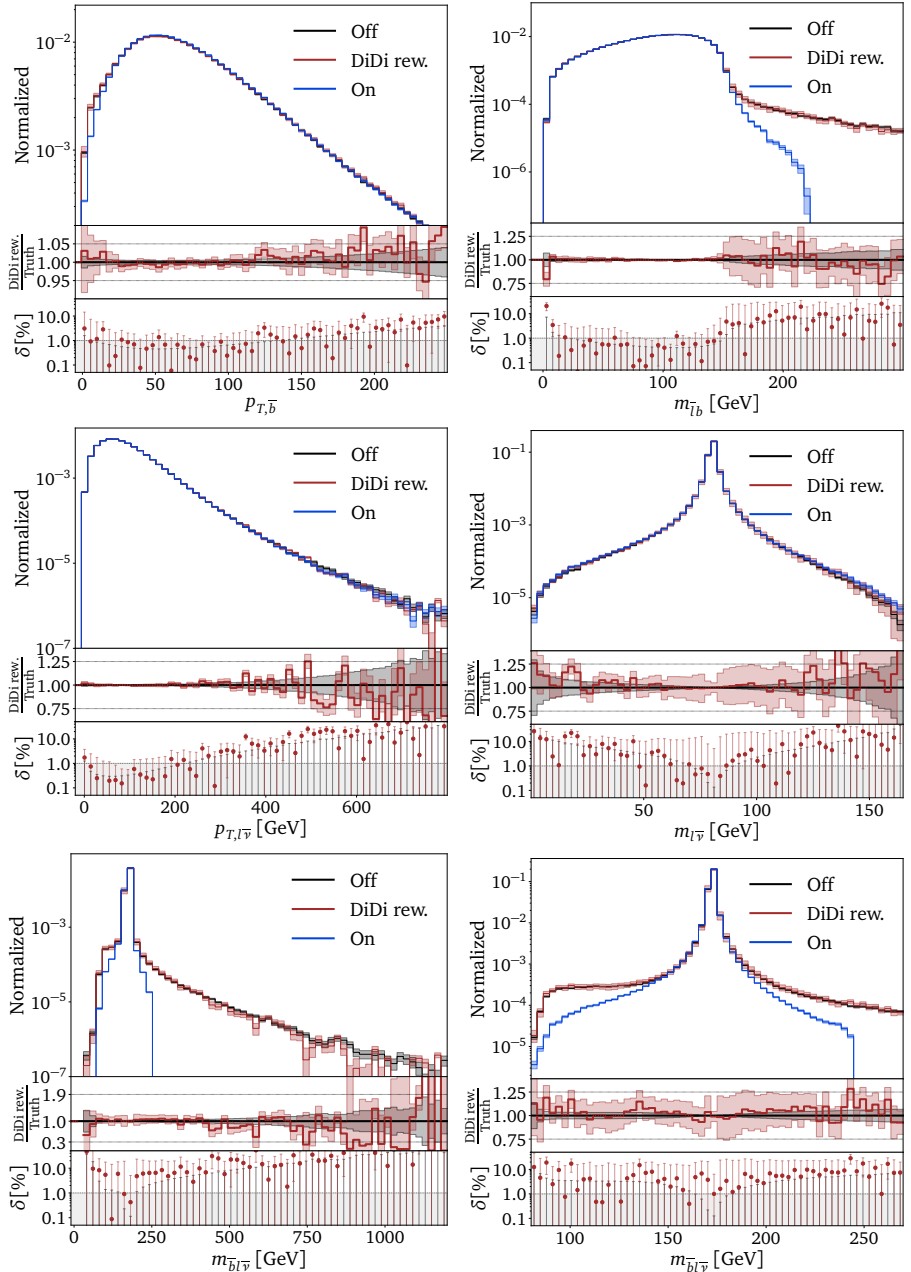

Figure 7: Results from our re-weighted Direct Diffusion off-shell generator, compared to the on-shell starting point and the off-shell target distributions.

For top pair production we have shown that already a relatively small network with limited training effort can reproduce the target off-shell distributions at the 10% level. Most importantly, the DiDi generation does not encounter any issues with learned features. The challenging on-shell peaks are described at the 5% level or better, as seen in Fig. 3.

Using a classifier reweighting we can improve its precision to the few per-cent level, even in challenging kinematic distributions, Fig. 7. Now the bulk precision exceeds 1%, and the precision in the extrapolation region is reliably below or at the 10% level. The same classifier allows us to control the performance of the generative networks and ensures that all local features in phase space are correctly learned. Finally, the Bayesian extension of the DiDi architecture provides us with a conservative uncertainty estimate, tracking the effects of statistically limited training data.

Our approach can be extended to higher-orders as long as the corresponding event samples can be provided as training data. Such an extension is not trivial because with each extra particle in the final state, e.g. due to an extra emission in the NLO real correction, the dimension of the phase space increases. We do not foresee any conceptual issues however, as increasing the dimension of the phase space in our setup is straightforward and relatively small event samples were needed at LO, and both diffusion networks as well as Classifiers have been shown to scale well with dimension. The size of the off-shell effect, or more specifically the ratio of the full and approximate off-shell prediction for a given observable, could in principle change dramatically when changing orders.[1] This is not a problem though, as DiDi is able to reproduce full off-shell predictions also in regions where the approximate off-shell prediction is vanishing. For realistic predictions we will also need to address issues related to matching to the parton shower, like shower starting scale and colourflow configurations not discussed here. For the latter we may have to include single top production in our on-shell data set.

Our fast generative surrogate is ready to be implemented standalone using Les Houches Event (LHE) format for input and output. Generative networks of this kind will also be part of the ML-enhanced ultrafast MADGRAPH event generator [23, 24].

## Acknowledgments

We thank Nathan Huetsch for many fruitful discussions.

**Funding information**   AB and TP are supported by the Deutsche Forschungsgemeinschaft (DFG, German Research Foundation) under grant 396021762 – TRR 257 *Particle Physics Phenomenology after the Higgs Discovery*. AB gratefully acknowledges the continuous support from LPNHE, CNRS/IN2P3, Sorbonne Université and Université de Paris Cité. Work at the University of Münster is supported by the BMBF through project *InterKIWWU* and by the DFG through SFB 1225 *Isoquant*, project-id 273811115, and the Research Training Group 2149 *Strong and Weak Interactions - from Hadrons to Dark Matter*. SPS is supported by the BMBF Junior Group *Generative Precision Networks for Particle Physics* (DLR 01IS22079). The authors acknowledge support by the state of Baden-Württemberg through bwHPC and the German Research Foundation (DFG) through grant no INST 39/963-1 FUGG (bwForCluster NEMO). This work was supported by the DFG under Germany's Excellence Strategy EXC 2181/1 - 390900948 *The Heidelberg STRUCTURES Excellence Cluster*.

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
