# Peer review of "Kicking it Off(-shell) with Direct Diffusion"

_SciPost Physics, doi:SciPost Phys. Core 7, 064 (2024)_

## Round 1 · Referee Report · Anonymous (Referee 3) · 2024-4-17

Strengths

1) Generic, novel and powerful method to transform high-dimensional distributions 2) No pairing of events needed 3) Very promising proof of concept for an example LHC application

Report

The criteria for publication are met.

The study opens a new pathway for fast surrogate modeling, with clear potential for multipronged follow-up work. In particular off-shell effects in dominant LHC background processes are modelled to high accuracy from the corresponding on-shell processes.

It provides a novel and synergetic link between particle physics and machine learning.

Requested changes

No further changes requested.

Recommendation

Publish (easily meets expectations and criteria for this Journal; among top 50%)

---

## Round 1 · Referee Report · Anonymous (Referee 2) · 2024-5-15

Strengths

A very interesting, innovative and important first step in the study of off-shell and finite width effects for processes relevant to LHC physics.

Weaknesses

The study is incomplete. The authors wrote that the method a priori is not limited to LO accuracy or top quark production and as such is more widely applicable. Beyond this, however, they have not demonstrated it at all. It is not enough to use the leading order to claim that this is sufficient evidence for the potential of the method, given the huge differences between the full off-shell effects at LO and NLO QCD or LO and NLO EW. In addition, the authors mentioned other methods in the literature, but did not make any comparison in terms of performance, speed, size of assigned error, etc.

Report

I would recommend that the article not be published in its current form.

Requested changes

The method should be tested on a few processes at NLO in QCD and a few comparisons should be made with methods already presented in the literature.

Recommendation

Ask for major revision

---

## Round 1 · Referee Report · Anonymous (Referee 1) · 2024-6-4

Strengths

Novel method for event generation of very complex processes

Weaknesses

Only leading order study

Report

I am actually quite happy with the response of the authors, as they clarified
many things for me. It turned out my main points of critic were based on some
missunderstandings on my side. Especially, regarding the included physical
effects and the generation of phase space dependent input variables, such as
colorflow and resonance histories.

I can also understand the authors point of view regarding the inclusion of
parton shower effects in the simulation. Nonetheless, I would recommend the
authors to perform the study including NLO QCD effects in the near future.

In summary, I can recommend the article for publication.

Recommendation

Publish (meets expectations and criteria for this Journal)

---

## Round 1 · Author Response

Answer to Report 1

Dear Referee,

Many thanks for a thorough consideration of our manuscript. We addressed all the issues that you raise and believe you will now find the manuscript suitable for a publication in SciPost. For more details please find our remarks/replies threaded into your report.

Sincerely, The authors

First, the authors seem to equate off-shell effects simply with kinematical edges in differential distributions. However, off-shell effects have an impact beyond the larger available phase space. For instance, the inclusion of top-quark decays provide spin correlations between the charged leptons that also have to be accurately described.

In our setup we find ourselves in the situation where the spin correlations are included exactly (for doubly resonant topologies at LO to be more precise) in the “on-shell” as well as the “off-shell” data set. In particular, the “on-shell” data include it thanks to the decays being modelled with the MadSpin method of Ref. [123]. Consequently the neural network does not need to worry about those. We added a note just before reference [123].

Furthermore, interference contributions between top-quark pair events and non-resonant continuum contributions can not be differentiated which will be necessary for accurate simulations.

The non-resonant contributions, as well as their interference with the resonant ones, are included in the “off-shell” data set. All the distributions that we investigated confirm that these contributions are correctly captured by our diffusion neural network + classifier setup. Note in particular the high tail of the m_lb distribution which is sensitive to both the single top contribution and its interference with tt~.

However, the authors fail to discuss these at all. Secondly, the motivation of the authors was to provide a method that could be employed to speed-up the event generation for future LHC simulations. The study presented here is performed with unweighted events at leading-order accuracy without taking into account further effects from parton showers. In consequence, the generative direct diffusion network is employed to generate hard events with off-shell kinematics, which still need to be interfaced with parton shower programs. To this end, additional information besides the momenta is necessary. Foremost, colorflow configurations and resonance histories need to be supplied. Providing the latter information solely based on momentum information is a challenging task and might diminish the accuracy of the predictions.

The colourflow configurations in our prediction can be trivially inherited from the “on-shell” data set. This is because in its default setup, the bb4l generator, used to generate the “off-shell” data set, does not consider colorflow configurations of the singly resonant contributions and the contribution due to non-resonant topologies, projected out using its default resonance history projectors, is suppressed by several orders of magnitude.

For example, in arXiv:1607.04538 it has been reported that the proper choice of resonance history is essential to obtain accurate predictions. For the same reasons, all events in our prediction can simply be attributed to the tt~ resonance history. Guessing from the final momentum configurations can lead to disastrous results.

According to Fig. 18 in arXiv:1607.04538, and the corresponding description, the resonance histories can be reconstructed from final momentum configurations very reliably.

The authors do not address these issues at all. At last, the precursor article (arXiv:2305.10465) already includes already a more complete study. In the previous article, hadronic Z-boson production up to 3 jets including parton shower effects, hadronization and multi-jet merging has been considered. It clearly, demonstrates that it is possible to push the direct diffusion method all the way to the end of the simulations. I wonder why the authors decided to work at the level of the hard scattering events instead of the 'final product' as it would have eliminated the problem of parton shower initial conditions completely.

Even though including parton showers in our data sets should be achievable, as demonstrated in other studies, we believe that in this particular setup it is actually undesirable. Firstly, the procedure of parton shower tuning is a crucial feature that needs to be preserved in order to make it possible for our predictions to profit from systematic improvements of the parton shower itself. Secondly, the parton shower is, in terms of computational effort, the easy task in a typical NLO+PS POWHEG setup, which is what we are eventually aiming for. Therefore, including it may complicate the neutral network’s task without providing an appreciable speedup.

Answer to Report 2 Dear Referee,

We are grateful for your feedback. It is indeed “... a very interesting and important study …”. We also appreciate your effort in listing various important issues that one might encounter when applying this method to top quark production and decay at NLO, even though it has no direct bearing for this study.

However, we respectfully disagree regarding the completeness of our study. There seems to be a fundamental misunderstanding regarding the main goal of this publication, which is to report on a novel method applied to a toy example of top quark production. Let us highlight that this method is a priori not limited to LO accuracy or to top quark production, and as such it holds broader interest. Moreover our idea is orthogonal to other applications of machine learning techniques in the context of assisting Monte Carlo event generation (such as arXiv:1808.07802, arXiv:2305.10475, arXiv:2311.01548 and references therein).

We agree that the next logical step would be to apply this method at NLO but such a step goes beyond the scope of this work.

We believe our reply addresses all issues that you raised, that are relevant to the actual content of our study, and that you will now find our manuscript suitable for publication in SciPost.

Sincerely, The authors

Dear Referee,

Thank you very much for your positive feedback regarding our study and recognizing the potential it holds. We would also thank you for your constructive remarks on what to change to clarify certain parts further. You can find a detailed reply to all of your points mentioned threaded into your report.

Sincerely, The authors

1) Optimal transport prescription is mentioned but details are missing

4) Fig. 4 caption "It illustrates the optimal transport algorithm chosen by the network training" - this needs to be explained.

We agree that it might have been misleading when we talked about the “optimal transport prescription” without explaining further what we mean by that. Our point is that the trained network has learned to transport each on-shell phase space point to a corresponding off-shell phase space point as is visible in Fig. 4. Here, we can see a clear linear relationship hinting that in terms of closeness, our model learned a meaningful transport map. However, it does so without enforcing any definition of closeness or optimality during training. We tried to clarify this statement more in the text and we hope that this will clear up any misunderstandings.

2) Two redundant degrees of freedom are said to increase the precision - the reader has difficulty understanding why

We understood the question and tried to clarify. However, it was more of an empirical finding than a theoretical one, meaning that in terms of explanation we can only offer speculations. We added a sentence in the text.

3) Define CFM the first time it is used

Thank you for pointing that out. We changed that.

5) Speed improvements are claimed but quantification is missing

In leading order speed is not a problem for on-shell or off-shell generation. So a comparison is not really sensible. However, including higher orders will change that drastically, so that our setup promises to speed up the generation of off-shell kinematics once higher order corrections are included. Here the speed difference might be decisive for LHC analyses.

6) Amortisation is mentioned but it is not clear how this will be used at scale: a given process will need training data; and it is not clear to what extent a training for one process can be reutilised for another process. The reader would appreciate a discussion of these issues to judge production readiness

We tried to not make any promises in the regard of amortisation and just mentioned that other processes can be treated similarly, meaning no amortisation but actually retraining the whole setup for that given process. A way to amortise the setup is left for future studies.

7) Add reference to related work such as https://arxiv.org/pdf/2312.10130.pdf

Unfortunately, this paper was published after our submission.

---

## Round 1 · List of Changes

1. Slight reformulation of the abstract.
  2. Specification of "CFM" in the last paragraph of the introduction (page 2).
  3. Updated description of event generation in Section 2 (page 3). Included more information on spin correlation and Bb4l settings
  4. Clarified why it is appropriate to restrict the setup to parton level (page 4)
  5. Clarified our description of the learned transport map (last paragraph page 6)
  6. Clarified why we include two redundant degrees of freedom during training (page 7)
  7. Changed caption of Figure 4

---

## Round 2 · Referee Report · Anonymous (Referee 2) · 2024-8-6

Report

The corrections made to the article go in the right direction and clarify the essence of the current study, while not exaggerating that the method used in the article can generate off-shell effects at higher-orders without any further modifications/problems. I can recommend the article for publication in its current form. However, I would like to encourage the authors to repeat the study at least at the NLO level in QCD.

Recommendation

Publish (meets expectations and criteria for this Journal)

---

## Round 2 · Author Response

Dear Editor,

Thank you so much for carefully considering the input from all the referees and determining the best next steps. We are also grateful for your valuable suggestions on how to improve our manuscript and to address the concerns raised by Referee No. 2. Please find replies/remarks to all of your points threaded into your letter.

Sincerely, The authors

Would it be possible to address the concerns of Referee No. 2 by discussing this point more explicitly and extensively in your paper, including emphasizing in the paper how your study is mainly about a novel application of ML ideas that is different from what done so far, We modified the abstract to make the goal of our study clear from the outset. Moreover, we fine tuned the introduction such that the emphasis on the novel ML idea is increased. and possibly also listing the extension to NLO QCD as part of the future developments? It would be helpful to maybe also add insights on what you expect to be the challenges involved and how you would plan to address them. We added a paragraph in the Outlook describing extensions to higher orders and the possible issues that may arise and how we’d address them.

---

## Editorial Decision

published